# Computer Simulation of Polyethylene Terephthalate Carbonated Beverage Bottle Bottom Structure Based on Manual–Automatic Double-Adjustment Optimization

**DOI:** 10.3390/polym14142845

**Published:** 2022-07-13

**Authors:** Shangjie Ge-Zhang, Xiaoli Chen, Haotong Zhu, Yuan Song, Yuyang Ding, Jingang Cui

**Affiliations:** 1College of Science, Northeast Forestry University, Harbin 150040, China; gzsj19@nefu.edu.cn (S.G.-Z.); cxl19@nefu.edu.cn (X.C.); 2019212493zht@nefu.edu.cn (H.Z.); sy2020222544@nefu.edu.cn (Y.S.); 2College of Foreign Languages, Northeast Forestry University, Harbin 150040, China; dyy2021@nefu.edu.cn

**Keywords:** computer modeling, PET bottle, carbonated drinks, stress cracking, simulations

## Abstract

PET bottlesare often used as airtight containers for filling carbonated drinks. Because carbonated drinks contain large volumes of CO_2_ gas, the container needs to bear a tremendous pressure from the inside of the bottle.If the stress exceeds the bearing limit, the material will show the phenomenon of local cracking and liquid overflow.For the structural design, the method of manual adjustment before automatic adjustment was adopted. First, through manual optimization, the initial optimal parameter combination was as follows:the inner diameter of the bottle bottom was 17 mm, the dip angle of the valley bottom was 81°, the deepest part of the valley bottom was 25 mm, and the outer diameter was 27 mm. Comsol software was used for automatic optimization. Compared with the original bottle bottom, the total maximum principal stress and total elastic strain energy in the bottle bottom after manual–automatic double optimization decreased by 69.4% and 40.0%, respectively, and the displacement caused by deformation decreased by 0.60 mm (74.1%). The extremely high reduction ratio was caused by manual–automatic double optimization.

## 1. Introduction

Polyethylene terephthalate (PET) is an ideal container for food and beverage packaging, with properties of non-toxicity, light weight, high transparency, and impact resistance. PET has been widely used in the packaging of carbonated beverages because it can block CO_2_ gas and retain the taste of soda water [1,2,3]. At present, the most commonly used carbonated beverage bottle was designed in the mid-1980s [4]. This claw-flap-shaped PET bottle with complex bottom structure was prone to stress cracking [5]. In order to solve the cracking problem of the PET bottle bottom, scholars have conducted a great deal of research on the process of PET bottle injection molding and the characteristics of the material itself. In terms of production technology, Lontos et al. [6] found that with the delay of pre-blowing pressure relative to the stretching effect, more materials would gather in the lower area of the final product by studying the influence of stretching and blow molding combination on the wall thickness distribution of 1.5Lt PET bottles. Akkurt et al. [7] used Taguchi method (TM), Grey Relational Analysis (GRA), and ECHIP to analyze the effects of preform temperature, stretching rod position, and final pressure, and optimized the performance parameters. In addition, the orientation and crystallinity of PET molecules are also the main factors that determine whether the mechanical properties of finished products are excellent [8,9,10]. Demirel et al. [11] used PET/nano hydroxyapatite (nHAp) composite particles to produce PET/nHAp bottles. The PET bottle blocked about 80% light transmission in a wavelength range of 400–700 nm and could be used in the food industry, especially in the packaging of dairy products, which are easily irradiated by light. Girard et al. [12] focused on the effect of annealing before stretching on strain-induced crystallization of PET and found that treatment at 120 °C for 60 s would lead to amorphous phase relaxation and higher crystallinity, and with time, higher mechanical strength and smaller shrinkage would be obtained. However, most of the above studies are about the characteristics of PET materials, without considering the geometric structure of products. In fact, the structural design of the bottle bottom is also one of the important factors affecting the stress cracking of the PET bottle bottom [13,14,15].

In this paper, by manually and automatically adjusting the structure of claw-flap PET carbonated beverage bottle bottoms, the cracking phenomenon of PET bottle bottoms due to stress concentration was effectively alleviated.Simple manual comparison is necessary, which can determine the optimal solution in a large range and reduce the number of operation groups in automatic optimization, thus, greatly shortening the operation time.Firstly, the variables were controlled manually for preliminary comparison and optimization. Based on the preliminary optimization, the computer was used for automatic calculation and optimization. The concrete steps were as follows: firstly, a single variable was adjusted by manual control and the best parameter combination was determined through the comparison of 10 groups of data. Then, by changing the material distribution, wall thickness, and fine structure shape, the combination parameters were automatically optimized on the premise of keeping the mass of consumable materials unchanged, further reducing the stress (23.4 MPa) and elastic strain energy (0.153 MPa cm^3^), and minimizing the deformation inthe bottle bottom (0.21 mm). This work can provide theoretical guidance for determining the mechanical structure parameters of PET carbonated beverage bottle bottoms. In the follow-up work, this project should explore the impact of the blow molding environment on the mechanical properties of bottle bottoms, which is combined with the actual production needs.

## 2. Principle and Experiment

### 2.1. Computer-Aided Modeling of Bottle Bottom Structure

SolidWorks simulation software was used to build the structure model of claw-flap PET carbonated beverage bottle bottoms.This model is composed of five concave claw-flap-like valleys, called the CF model. The bottom model was equally divided into five identical 72° regions, so only one 72° region was needed. The complete bottom was generated by copying and rotating the 72° region with an rotating array component. In this paper, in order to compare and only compare the performance of the structure model before and after optimization. In the whole optimization process, the influence of other environmental factors, such as the temperature and time of blow molding, was not increased. In the preliminary manual model optimization, the research did not consider the thickness change and thickness distribution for the time being, but only compared the influence of the geometric structure of the model on the mechanical properties of the bottle bottom, so a uniform bottle bottom thickness of 1 mm was assumed first. It is worth noting that the claw-flap-shaped bottle bottom structure is an irregular geometric shape. In SolidWorks modeling, if the thickness of all irregular interfaces is defined as 1 mm, then there will be obvious cracks or cliffs at the intersection of the two interfaces due to non-coincidence, and the bottle bottom cannot be regarded as an unbroken whole. The modeling used SolidWorks’ own shell-pulling command to make all parts smoothly connected; there was no cliff, and the average value of the final result was still 1 mm.

As shown in Figure 1, this article named several important parts of the model, which are the inner diameter of the bottle bottom (the Aidentification inthegraph), the inclination angle of the valley (B), the deepest part of the valley bottom (C), and the outer diameter (D). Parameter change in manual optimization was also carried out in SolidWorks. By changing the parameters, the combination of 10 groups of single variables would lead us to preliminarily determine the optimal solution of these four parameters. The parameters at A and D are changed by directly changing the diameter of the circle in the sketch, the draft angle in cutting and stretching is used at B, and the distance from the datum plane to xoy plane is actually at C.

### 2.2. Selections of Material and Finite Element Mesh

The model was brought into Comsol simulation software and the bottle bottom material was defined as linear thermoplastic PET with a thickness of 1 mm. The material density of the plastic was 1190 kg m^−3^, and its Young’s modulus, Poisson’s ratio, and thermal expansion coefficient were 3200 MPa, 0.35, and 7 × 10^−5^ K^−1^, respectively. In the mesh selection, triangular mesh is used to divide the model, because triangular mesh has good boundary adaptability, which can prevent the generation of bad elements and make the calculation converge quickly. Polygonal elements with more sides (such as quadrilateral elements) have higher accuracy than triangular elements, so the grid number of triangular elements was appropriately increased to make up for the lack of accuracy, totaling 45,230 elements. No more elements were introduced because too small meshes would increase the processing time and might even lead to non-physical understanding.

### 2.3. Pressure Setting

It is necessary to apply constant pressure to the bottle bottom to obtain the maximum principal stress, elastic strain energy, and deformation of finite element simulation analysis. Under the same pressure, by comparing the results of the maximum principal stress, we can easily obtain the advantages and disadvantages of each parameter combination set manually, which is also applicable in automatic optimization. This article chose the pressure of carbonated beverage bottle at 20 °C with 8.45 g L^−^^1^ CO_2_ in the bottle; that is, a pressure of 0.4 MPa (0.04 kg mm^−2^), which simulated the real situation of the pressure caused by the common CO_2_ content in carbonated beverage bottles. The direction of the force was outward along the normal of the bottle bottom.

### 2.4. Constraint Conditions

Considering that there are some constraints in the actual production process, such as sudden sharp protrusions or depressions, that are not allowed, before stress testing and shape optimization, some necessary constraints need to be added to make the results conform to the reality, and the optimized model can be applied to product production.

#### 2.4.1. Fixed Edge

The fixed edges of the constraint were set as the inner and outer edges of the upper edge to prevent the rigid body from translating and rotating. The upper edge was chosen because the research object was the bottle bottom, not the bottle body, and the fixed edge was the extension of the bottle bottom and connected with the bottle body. Constraint Equation 1 is presented by directly setting a fixed edge:(1)d=0

Before optimization, the fixed edges were added with fixed constraint and prescribed displacement.

#### 2.4.2. Free Shape Domain

Although the Yeoh model [16] does not perform well in dealing with the complex strain state of large deformation, it is applicable and accurate for the small deformation in this study. The nonlinear Yeoh smoothing type was used to determine the mesh deformation, which could describe a wide deformation range. Compared with other methods [17,18,19,20], Yeoh smoothing prevents the further deformation of these regions to some extent, and effectively distributes the mesh deformation in the domain more evenly. Its strain energy expression is as follows:(2)$$W=C_{1}\left(I_{1}-3\right)+C_{2}{\left(I_{1}-3\right)}^{2}+C_{3}{\left(I_{1}-3\right)}^{3}$$

Among them, the value of stiffing factor $C_{2}$ controls the nonlinear hardening of artificial materials under deformation, which is set to 10 in this experiment, and $I_{1}$ is Cauchy–Green deformation tensor.

#### 2.4.3. Free Shape Boundary

A free shape boundary was added to the edge of the free shape domain. Simply put, the points on the boundary are limited to move in the area of $\pm d_{max}$. The constraint equation of free shape boundary is as follows:(3)$$d=c+R_{min}^{2}\nabla_{\|}^{2}d,\;\;\;\;\;\;-d_{max}\leq c_{i}\leq d_{max}$$
where $d_{max}=1$ mm is the set maximum allowable displacementand $R_{min}$ is the filtering radius. The setting of free shape boundary ensured that there was no sudden sharp protrusion or depression at a certain point similar to a sea urchin shape.

### 2.5. Steady State Solver

Considering that the matrix of the model was sparse, the PARDISO direct solver [21,22,23] based on LU decomposition was adopted for the steady-state solution. This solver is located in Intel Mathematical Kernel Library (MKL), which has very good computational efficiency and parallelism. With an increase in the number of computing nodes, PARDISO has a nearly linear acceleration ratio. Although the direct solver uses more memory than the iterative solver, it is robust.

### 2.6. Data Comparison

In this paper, the steadystates of 10 groups of modulated models are calculated, respectively, and the optimal combination in the manual group was determinedby comparing the Von Mises stress. In the subsequent iterative process of automatic optimization, Von Mises stress, elastic strain energy, and deformation were also calculated seriatim and used as important indexes to evaluate the mechanical structure performance of the bottle bottom [24,25,26]. The final comparison covers four types: the original model (O-CF bottom), the model only optimized manually (M-CF bottom), the model only optimized automatically (A-CF bottom), and the model with manual–automatic double optimization (MA-CF bottom).

#### 2.6.1. Von Mises Stress and Elastic Strain Energy

Von Mises stress is an equivalent stress based on shear strain energyand the standard unit should be Pascal (Pa), while the most suitable unit size of bottle bottom stress is mega-Pascal (MPa). It comprehensively considers the first, second, and third principal stresses, and its value is:(4)$$\sigma_{s}=\sqrt{\frac{{\left(\sigma_{1}-\sigma_{2}\right)}^{2}+{\left(\sigma_{2}-\sigma_{3}\right)}^{2}+{\left(\sigma_{3}-\sigma_{1}\right)}^{2}}{2}}$$
where $\sigma_{1}$, $\sigma_{2}$, and $\sigma_{3}$ refer to the first, second, and third principal stresses, respectively. Elastic strain energy is the energy stored by a solid due to deformation under the action of external force and its unit is Joule (J). In this paper, MPa cm^3^ was used as the homogenization unit.

#### 2.6.2. Deformation

Deformation is divided into elastic and plastic deformation. In this chapter, the elastic deformation is discussed; that is, the relative position of each point is changed by the external force, and when the external force is removed, the solid returns to its original state. The principle is simple Hooke’s law:(5)F=kx
where k is a constant and the elastic coefficient of an object and x is the deformation.

### 2.7. Optimization

#### 2.7.1. Optimized Content

When the stress concentration caused by the structure of the bottom exceeds its maximum value, the bottle bottom will crack. In addition, when the bottle with high internal pressure is opened, the elastic strain energy stored in the deformed bottle bottom will be released, meaning that the liquid in the bottle overflows. After manually determining the data combination in a large range, automatic optimization focused on adjusting the thickness and material distribution of the bottle bottom. Theoretically, the more materials invested in the production of a bottle bottom, the greater the overall average thickness of the bottle bottom, and the less likely it is to crack. However, considering the cost and other factors in actual production, the research took the premise that the total quality of the materials used was the same to carry out automatic optimization, which was feasible and meaningful.Therefore, in automatic optimization, the total mass of materials used should be the same. Through shape optimization, the thickness and material distribution of the bottle bottom were changed to solve the minimum and maximum principal stress and total elastic strain energy, which cannot be completed by manual optimization.

#### 2.7.2. Optimization Method

The shape optimization method was used to change the small geometric parameters, thickness, and material distribution parameters in the model, so as to automatically optimize the bottle bottom. The solution method is a method of moving asymptotes (MMA) [27], which is a three-level nonlinear algorithm based on gradient and belongs to a convex programming method. By introducing moving asymptote, it transforms the implicit optimization problem into a series of explicit, separable, simpler, and strictly convex approximate sub-optimization problems. In each iteration, the gradient algorithm can be used to solve the convex approximation sub-optimization problem to obtain new design variables. This optimization used the globally convergent version of MMA (GCMMA) [28], including external iteration and internal iteration. The subproblems of the k-th internal iteration under the *v*-th external iteration are as follows:(6)$$\mathrm{minimize}{\tilde{f}}_{0}\left(x\right)+z+\frac{1}{2}d_{0}z^{2}+\overset{m}{\underset{i=1}{\sum }}\left(c_{i}y_{i}+\frac{1}{2}y_{i}^{2}\right)$$
(7)$$\mathrm{subject}\;\mathrm{to}{\tilde{f}}_{i}\left(x\right)-a_{i}z-y_{i}\;\leq \;b_{i},i=1,\ldots ,m$$
(8)$$\alpha_{j}\leq x_{j}\leq \beta_{j},\;j=1,\ldots ,n$$
(9)$$y_{i}\geq 0,\;i=1,\ldots ,m$$
(10)z≥0

The approximate functionsare constructed as:(11)$${\tilde{f}}_{i}\left(x\right)=\overset{n}{\underset{j=1}{\sum }}{\tilde{f}}_{ij}\left(x_{j}\right)=\overset{n}{\underset{j=1}{\sum }}\left(\frac{p_{ij}}{u_{j}-x_{j}}+\frac{q_{ij}}{x_{j}-l_{j}}\right),\;i=0,1,\ldots ,m.$$

Between each outer iteration, the bounds ($\alpha_{j}$, $\beta_{j}$) and the asymptotes ($l_{j}$, $u_{j}$) are updated. In this paper, the number of iterations of automatic optimization was 17; that is, there was no obvious change in the value of 15–17 iterations, so it was judged that the model converges and terminates.In addition, the boundary conditions of the mathematical model are:(12)$$\mathrm{minimize}f_{0}\left(x\right)+z+\overset{m}{\underset{i=1}{\sum }}\left(c_{i}y_{i}+\frac{1}{2}y_{i}^{2}\right)$$
(13)$$\mathrm{subject}\;\mathrm{to}f_{i}\left(x\right)-a_{i}z-y_{i}\;\leq \;\;f_{i}^{max},i=1,\ldots ,m$$
(14)$$x_{j}^{min}\leq x_{j}\leq x_{j}^{max},\;j=1,\ldots ,n$$
(15)$$y_{i}\geq 0,\;i=1,\ldots ,m$$
(16)z≥0

In the above formulas, $f_{0},\;f_{1},\;\ldots ,\;f_{m}$ are given differentiable functions, while $x_{j}^{min},\;x_{j}^{max},\;a_{i},\;c_{i}$ are given real numbers and satisfy $x_{j}^{min}\leq x_{j}^{max}$, $a_{i}\geq 0$ and $c_{i}\geq 0$.

## 3. Results and Discussion

### 3.1. Comparison of Model Structure before and after Optimization

Figure 2 shows a comparison of the model structure before and after optimization; that is, a schematic diagram of the micro-geometric parameters, thickness, and material distribution of the optimized model. The range of 0–1 in the legend is the boundary displacement relative to the normal direction. The lowest value is 0, indicating no movement, and the highest value is 1, indicating a thickness change of 1mm relative to the normal direction. It can be seen that the material with less stress is mainly moved to the junction of the outer diameter and the bottom of the groove, which is also where the stress of the O-CF bottom is the largest.The location of the maximum principal stress is the same as that in similar articles [29,30], which proves the correctness of the experiment.The optimization results are repeatable, and the results of multiple optimization solutions of this model are only slightly below the relative tolerance; the relative tolerance is limited to 10^−3^. The results of repeated experiments verify this view.

### 3.2. Surface Von Mises Stress

The surface Von Mises stress distribution before and after optimization is shown in Figure 3, and more detailed parameters are given in Table 1. No matter the CF model before and after optimization, its maximum principal stress is located in the outer diameter of the bottle bottom, especially at the junction of the outer diameter and the groove bottom, there is obvious stress concentration. The maximum surface stresses of the O-CF bottom, M-CF bottom, and MA-CF bottom are 72.8 MPa, 66.9 MPa, and 23.4 MPa, respectively. After manual optimization and manual–automatic optimization, the reduction rates reach 8.10% and 67.86%. In addition, the A-CF bottomis also compared, which means that the automatic optimization is performed directly and the manual steps are skipped. The maximum stress on the surface of the A-CF bottom is 24.1 MPa (66.90%) and compared with the 17 iterations of the MA-CF bottom, the number of iterations of the A-CF bottom is increased to 20, which increases the amount of calculation and time. The distribution of maximum principal stress in the computer simulation of bottle bottom cracking by the O-CF bottom is basically consistent with the cracking phenomenon in the surface stress cracking test, which proves that the maximum principal stress plays a key role in the cracking process [29,30,31,32]. In addition, it should be pointed out that the maximum stress of the material sometimes appears inside the object, and Figure 3 is added to visualize the stress distribution. In order to express the maximum stress more accurately, the research introduces the concept of overallVon Mises stress, and the data can be seen in Table 1.

Compared with similar articles, the maximum principal stress on the surface of the bottle bottom after optimization decreased by 21% and 10.8%, respectively [30,33]. Obviously, the reduction ratio of 67.86% in this paper achieved better results.

This study, therefore, provides a new strategy to optimize the model structure.Our results prove that the double-optimization results are superior to the other two optimization methods. This is because simple manual optimization can adjust the given parameters greatly, but cannot realize the synergistic influence among the parameters. Compared with simple manual optimization, simple automatic shape optimization is undoubtedly more accurate, but it has a large amount of calculation and many iterations (20 times). Moreover, automatic optimization is also aimed at finetuning. For the overall specific parameters, especially the dip angle of the valley bottom, it is difficult to change the angle by automatic optimization without manual adjustment.

### 3.3. Overall Von Mises Stress

It can be seen from Table 1 that compared with the O-CF bottom (77.9 MPa), the overall maximum principal stress inthe M-CF bottom model is 69.2 MPa, which is reduced by 8.7 MPa with a reduction rate of 11.2%; after automatic optimization, the overall maximum principal stress in the MA-CF bottom model is 23.8 MPa, and the reduction rate reaches 69.4%. This proves that in the aspect of optimization, it is difficult to manually change each factor, and it is easy to cause conflicts among variables. The model cannot achieve the best optimization, so it is suitable for preliminary judgment. Automatic optimization, which can take all kinds of factors into account, presents much better results and can maintain higher accuracy.

It is worth noting that the overall Von Mises stress given in Table 1 is, indeed, higher than the corresponding surface Von Mises stress in Figure 3, because the maximum stress is located inside the bottle bottom instead of the surface, but the stress relationship is still corresponding. In addition, the minimum stress inthe MA-CF bottom model is higher than that of the M-CF bottom model, which is due to the fact that the mass of consumable materials is unchanged, and the materials in the area with low stress are appropriated in the process of automatic optimization, which changes the structure of the area and reduces the thickness of the area, but the bottom fracture is caused by the maximum stress instead of the minimum stress. Therefore, a moderate increase in the minimum stress is acceptable.

### 3.4. Total Elastic Strain Energy

In this paper, it was considered that the model tended to converge when the relative tolerance of the iterative solver was less than 10^−3^. Figure 4 shows the total elastic strain energy-iteration number diagram forthe MA-CF bottom, with 17 iterations. In addition to the maximum principal stress that caused the cracking of the bottle bottom in the previous section, this research also considered the total elastic strain energy of the bottle bottom, which was shown for a pleasurable drinking experience after actual sales. When the sealed container is opened, the pressure greater than atmospheric pressure brought by CO_2_ gas in the bottle is released, and the elastic strain at the bottom of the bottle can make the liquid overflow. Too-high elastic strain energy will make the liquid overflow more, resulting in an uncomfortable drinking experience. It can be seen that the elastic strain energy inthe M-CF bottom is reduced from 0.255 MPa cm^3^ to 0.236 MPa cm^3^ by 7.45% after the first manual optimization, and that of the MA-CF bottom is reduced to 0.153 MPa cm^3^ by 40.0% after the second automatic optimization. This study provides a new research direction for the introduction of elastic strain energy at the bottom of the bottle, which combines the needs of science and public life, and this is the significance of science.

### 3.5. Material Deformation

The bottom of the bottle will be deformed under pressure. If the pressure is excessive and the deformation exceeds the bearing limit of the material itself, the material will break, and the macroscopic manifestation is the cracking of the bottom of the bottle. In addition, inordinate deformation will cause instability when placing the bottle. A pressure of 0.4 MPa was applied to the model before and after optimization.The obtained material deformation results are shown in Figure 5. It can be clearly seen that the material deformation mainly occurs in the center of the bottle bottom, diverges outward, and gradually decreases. Before and after the first manual optimization, the maximum deformation inthe M-CF bottom model decreased by 0.25 mm, with a reduction rate of 30.9%; after the second optimization, the deformation inthe MA-CF bottom was only 0.21 mm, which was reduced by 74.1%. The reduction in deformation also shows the structural stability of the bottle bottom, and the optimization is successful.

Generally speaking, this study established an evaluation covering the maximum principal stress, elastic strain energy, and deformation, of which elastic strain energy and deformation were not studied in detail in previous articles. This improves the evaluation system of PET bottle bottom performance and helps people engaged in plastic packaging to get more valuable information. However, this research also has limitations. Microscopically, we did not consider the influence of molecular orientation and crystallinity on the fracture characteristics of the bottle bottom. Secondly, the temperature is necessary during the bottle processing, so it is necessary to further consider the heat transfer during the processing. Finally, according to statistics, the high temperature in summer will make the bottom of the bottle break more easily. Therefore, after the bottle is produced and filled with a beverage, further research should also consider the influence of pressure change and softening in the bottle on the performance of the bottle at high temperature.

## 4. Conclusions

In this paper, the influence of geometric structure on the cracking of a PET bottle bottom was analyzed and optimized. Manual–automatic double adjustment was adopted to optimize the bottom structure. Compared with the original bottom, the total maximum principal stress and total elastic strain energy in the optimized bottle bottomdecreased by 69.4% and 40.0%, respectively, and the displacement caused by deformation decreased by 0.60 mm (74.1%). The advantages of the double-adjustment method of automatic optimization after manual adjustment are:The characteristics of high precision inautomatic optimization are maintained;Manual preprocessing saves computing resources.

## Figures and Tables

**Figure 1 polymers-14-02845-f001:**
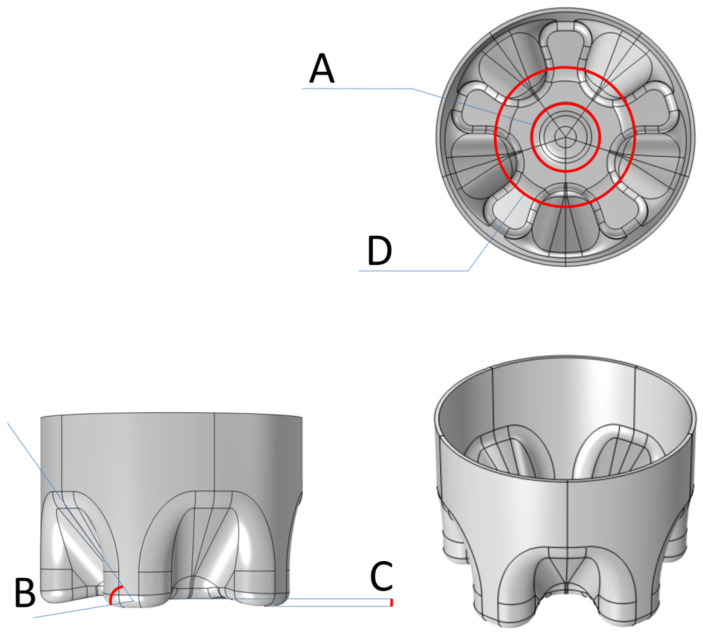
Schematic diagram of CF model and naming: inner diameter (**A**), inclination angle of the valley (**B**), deepest part of valleybottom (**C**) and outer diameter (**D**).

**Figure 2 polymers-14-02845-f002:**
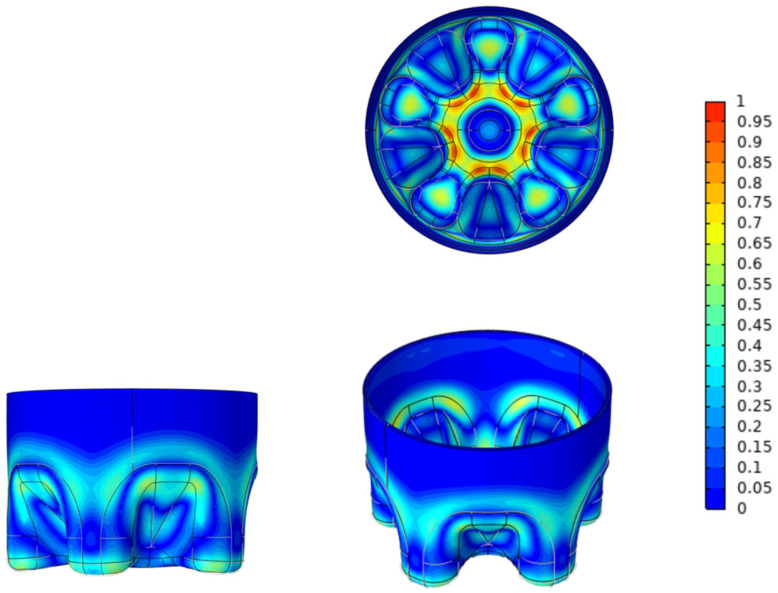
Comparison of the model structure before and after optimization.

**Figure 3 polymers-14-02845-f003:**
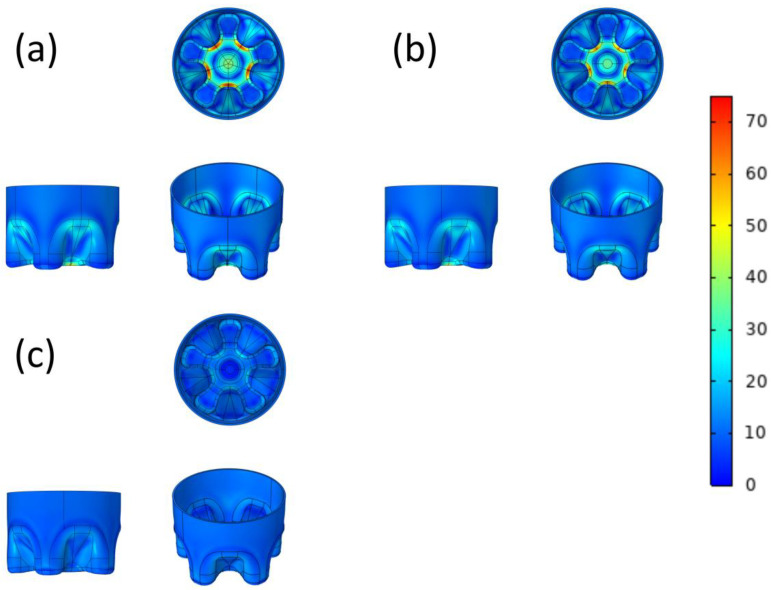
The surface Von Mises stress (MPa) distribution of O-CF bottom (**a**), M-CF bottom, (**b**) and MA-CF bottom (**c**).

**Figure 4 polymers-14-02845-f004:**
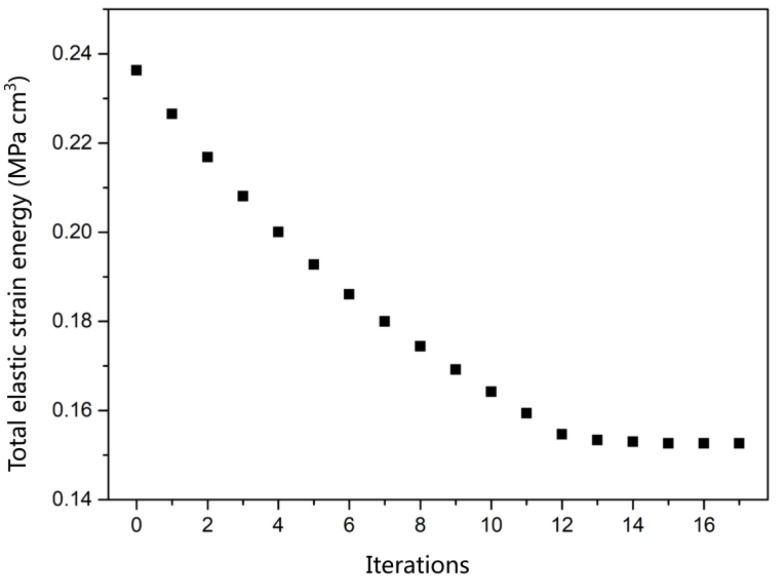
Total elastic strain energy (MPa cm^3^)-iteration diagram of MA-CF bottom.

**Figure 5 polymers-14-02845-f005:**
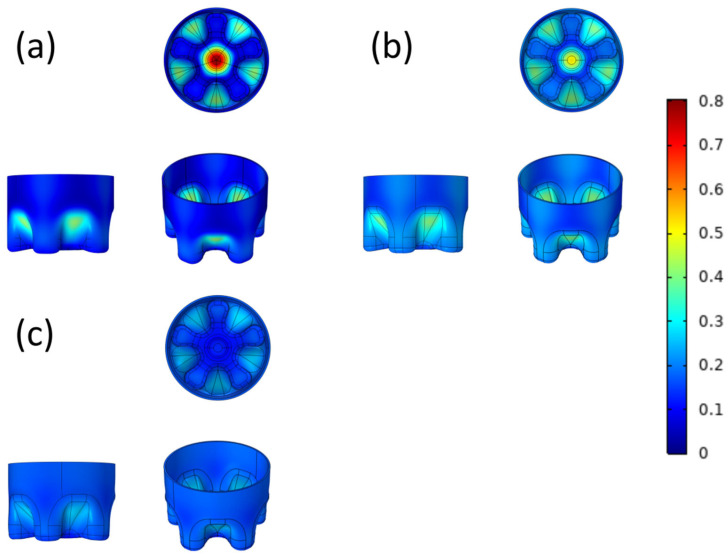
The stress deformation value (mm) of O-CF bottom (**a**), M-CF bottom (**b**), and MA-CF bottom (**c**).

**Table 1 polymers-14-02845-t001:** The maximum values of calculation results under different optimization degrees inthe CF model.

Degree of Model Optimization	Surface Von Mises Stress	Overall Von Mises Stress	Elastic Strain Energy	Deformation
O-CF bottom	72.8 MPa	77.9 MPa	0.255 MPa cm^3^	0.81 mm
M-CF bottom	66.9 MPa	69.2 MPa	0.236 MPa cm^3^	0.56 mm
MA-CF bottom	23.4 MPa	23.8 MPa	0.153 MPa cm^3^	0.21 mm

## Data Availability

Not applicable.

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
