# Peer review of "Computer Simulation of Polyethylene Terephthalate Carbonated Beverage Bottle Bottom Structure Based on Manual–Automatic Double-Adjustment Optimization"

_polymers, 2022, doi:10.3390/polym14142845_

Round 1

Reviewer 1 Report

The proposed manuscript is interesting; there are many weaknesses that need to be improved. This based on the following:

·        It is recommended not to use acronyms in the title without first defining them

·        Line 9-27: The abstract should be reviewed again because it is very general and the objective is not clear.

·        The scope of the study is not well defined, the authors could better express it in the abstract

·        Introduction:

o   some paragraphs of the introduction should be shorter

o   Authors should define some acronyms like "PET"

o   The objective should be integrated at the end of the introduction

·        What are the boundary conditions of the mathematical model that is used?

·        Line 179: Authors must write in the third person

·        In section 2.6.1. von Mises, correct Von Mises

·        Line 195: Deformation is divided into elastic deformation and plastic deformation; fix: Deformation is divided into elastic and plastic deformation

·        It is necessary to review the equation 11

·        In section 3.2. Surface von Mises, correct Surface Von Mises

·        Correct throughout the document: Von Mises

·        In figure 4, the units should be MPa or Joules, define one of the two

·        The discussion of results should be enriched, only the authors describe the results but they are not discussed.

·        it is necessary to indicate the repeatability of the tests

·        The authors present very extensive paragraphs in the conclusions, they must be specific. This section needs to be revisited

·         The authors present 32 references, Review the format of the journal, for example journals must be abbreviated

Author Response

Thank you for your kind and quick comment. Please see the attachment.

Reviewer 2 Report

The submitted manuscript is necessary for liquid packaging container design. The people working in plastic packaging can get valuable information from here. The selective comments towards the authors for revisions are the following:

[1]  In the Results and discussion part, section 3.2 to 3.5 needs to cite supporting references. It will help the reader to compare similar published data.

[2]  The conclusions section should be concise and informative for the readers. Some of the lines in this paragraph are too large to keep attention. The authors have to simplify the complex sentences.

Author Response

(The authors gave the same response as above.)

Round 2

Reviewer 1 Report

The authors have made the changes suggested in the manuscript, further discussion of results is needed, there is only a description. It is suggested to homogenize units, all in MPa or Joules. The article can be accepted for publication after making the discussion.

Author Response

(The authors gave the same response as above.)
